# Nutrition Strategies Promoting Healthy Aging: From Improvement of Cardiovascular and Brain Health to Prevention of Age-Associated Diseases

**DOI:** 10.3390/nu15010047

**Published:** 2022-12-22

**Authors:** Monika Fekete, Zsofia Szarvas, Vince Fazekas-Pongor, Agnes Feher, Tamas Csipo, Judit Forrai, Norbert Dosa, Anna Peterfi, Andrea Lehoczki, Stefano Tarantini, Janos Tamas Varga

**Affiliations:** 1Department of Public Health, Faculty of Medicine, Semmelweis University, 1089 Budapest, Hungary; 2Department of Haematology and Stem Cell Transplantation, National Institute for Haematology and Infectious Diseases, South Pest Central Hospital, 1097 Budapest, Hungary; 3Department of Neurosurgery, The University of Oklahoma Health Sciences Center, Oklahoma City, OK 73117, USA; 4Department of Health Promotion Sciences, College of Public Health, The University of Oklahoma Health Sciences Center, Oklahoma City, OK 73117, USA; 5Peggy and Charles Stephenson Oklahoma Cancer Center, Oklahoma City, OK 73104, USA; 6Department of Pulmonology, Semmelweis University, 1083 Budapest, Hungary

**Keywords:** nutrition, aging, healthy aging, prevention, geriatric diseases

## Abstract

Background: An increasing number of studies suggest that diet plays an important role in regulating aging processes and modulates the development of the most important age-related diseases. Objective: The aim of this review is to provide an overview of the relationship between nutrition and critical age-associated diseases. Methods: A literature review was conducted to survey recent pre-clinical and clinical findings related to the role of nutritional factors in modulation of fundamental cellular and molecular mechanisms of aging and their role in prevention of the genesis of the diseases of aging. Results: Studies show that the development of cardiovascular and cerebrovascular diseases, neurodegenerative diseases, cognitive impairment and dementia can be slowed down or prevented by certain diets with anti-aging action. The protective effects of diets, at least in part, may be mediated by their beneficial macro- (protein, fat, carbohydrate) and micronutrient (vitamins, minerals) composition. Conclusions: Certain diets, such as the Mediterranean diet, may play a significant role in healthy aging by preventing the onset of certain diseases and by improving the aging process itself. This latter can be strengthened by incorporating fasting elements into the diet. As dietary recommendations change with age, this should be taken into consideration as well, when developing a diet tailored to the needs of elderly individuals. Future and ongoing clinical studies on complex anti-aging dietary interventions translating the results of preclinical investigations are expected to lead to novel nutritional guidelines for older adults in the near future.

## 1. Introduction

The aging of the population of the Western world represents one of the greatest challenges for sustainable development. At present, approximately 9% of the European population is older than 65. This is projected to increase to 25% by 2050. Morbidity and mortality continuously increase with age [1]. There are several conditions that affect an older adult’s life, such as sensory decline (decrease or loss of hearing, poor eyesight), cardiovascular diseases, diabetes mellitus, depression, dementia, sarcopenia [2,3,4,5,6,7,8,9], gait and balance abnormalities and falls [10,11,12,13,14,15,16,17,18,19,20], and chronic obstructive pulmonary disease [21]. These diseases share several risk factors. One of them is an unhealthy diet accompanied by diseases related to a poor diet [22], such as overweight, obesity [23] and metabolic syndrome, which may affect up to 75% of the population aged over 60 [24]. Preclinical studies provide ample evidence that several components of diet, such as protein, carbohydrate and fat content as well as intake of calories, play important roles in regulating aging processes and longevity and development of age-associated diseases, including cardiovascular and cerebrovascular diseases, cognitive impairment and dementia, and malignant diseases [19,25,26,27,28,29,30,31,32,33,34,35,36,37,38,39,40,41,42,43,44,45,46,47,48,49,50,51,52,53]. Dietary regimens that favor reduced calorie intake were shown to delay aging and the genesis of age-associated diseases [31,47]. As the proportion of the elderly is on the rise, which is associated with an increased burden of disease and expenditure, evidence-based preventive policies and health promotion programs to promote healthy aging are needed to ensure the future functioning of social health care and pension systems. Healthy diets with anti-aging potential [54,55] are essential to prevent the development of chronic diseases and to promote healthy aging [56]. The aim of the present review is to summarize the role of diets in the development of common age-related chronic conditions (cardio- and cerebrovascular disease, neurodegeneration and neuropsychiatric disorders), and to provide an overview of their role in modulating the aging process itself.

## 2. Cellular and Molecular Mechanisms of Aging: Regulation by Nutrition and Diets

Geroscience research has identified basic biological mechanisms driving aging [57]. These mechanisms include pathways that drive organismal aging according to a biological timetable. These processes are regulated by changes in gene expression and epigenetic regulatory mechanisms [58,59,60] and are coordinated by endocrine pathways, such as the insulin/insulin-like growth factor-1 (IGF-1) signaling pathway [61,62]. There are multiple evolutionarily conserved nutrient sensing pathways, whose activity is regulated by nutrient levels and which regulate cellular metabolism and also influence aging and determine survival and lifespan (IGF-1, mammalian target of rapamycin (mTOR), sirtuins and AMP-activated protein kinase (AMPK) [30,42,63,64,65,66,67,68,69,70,71,72]. Additionally, certain aging phenotypes are also determined by stochastic processes, including macromolecular damage [73], which leads to the gradual “wearing and tearing” of cells and tissues [74]. Part of these age-related changes are driven by increased production of oxygen free radicals [75,76,77,78,79,80,81,82,83] and by the accumulation of cross-linked proteins [84,85] and DNA damage [86,87,88,89,90]. Some of this damage accumulates over time, while some is repaired by different repair mechanisms [91,92]. Importantly, both programmed mechanisms of aging and stochastic processes contributing to the genesis aging phenotypes are impacted by nutrition and diets (e.g., caloric restriction confers multifaceted anti-aging effects) [93,94]. There is strong evidence that unhealthy diets (e.g., a high fat diet, Western diets, and methionine-rich diets), because of accelerated cellular aging, exacerbate the development of age-related diseases and shorten lifespan in laboratory animals [95,96,97,98,99,100,101,102,103,104]. Development of aging phenotypes and the pathogenesis of age-related diseases are also influenced by a heightened state of low-grade sterile systemic inflammation (“inflamm-aging”) [105]. Recent findings highlight the association between inflammation and deleterious frailty status in older adults [106,107]. Inadequate diets (e.g., a high fat diet [95,96,97,98,99], diabetogenic diets, high methionine diets [108,109,110]) are an important contributing factor to this low-grade systemic inflammation, but it is also one of the easiest modifiable interventions for elderly individuals to intervene on the process of aging [105]. Preclinical studies have developed a wide range of dietary interventions to delay aging and prevent development of age-related diseases in laboratory animals. These include caloric restriction, methionine restriction, diets enriched with polyphenols, time-restricted feeding and intermittent fasting regimens. The results of preclinical research into the role of dietary factors in regulation of aging processes have been incorporated in the dietary recommendations developed for older adults. These findings, taken together with the results of epidemiological studies, led to the clinical development of various dietary patterns and interventions, including the Okinawan diet, Mediterranean diet, caloric restriction and intermittent fasting regimens, and high-carbohydrate-low-protein and ketogenic diets to combat aging in humans.

## 3. Dietary Factors Influencing Mortality

### 3.1. Nutritional Recommendations for the Elderly

Nearly 50% of premature deaths are linked to lifestyle, such as our diet [111]. A diet can be characterized quantitatively, by its calorie content, or qualitatively, for example, by its macro- and micronutrient content.

The amount of required daily calorie intake changes with age [112]. The average energy need of people over 60 depends on their anthropometric parameters and their daily physical activity [113]. A male with a sedentary lifestyle between 60 and 69 requires around 2000 calories a day, while above age 70 the required amount of 1900 calories is somewhat lower than that. In the case of an active lifestyle, these thresholds increase to 2600–2900 and 2500–2700 a day, respectively. In the case of an active life, females between 60 and 69 are advised to consume 2100–2300 calories a day, while females over 70 years of age should consume 2000 to 2200 calories daily [114].

Apart from the daily calorie intake, the macronutrient composition of the diet is also a key factor [115,116]. New evidence shows that the response to dietary fat intake may be based on individual circumstances and the rise in low-density lipoprotein (LDL) cholesterol caused by saturated fats may represent a normal rather than a pathologic response, with different factors, such as gut microbiota, mediating the response [117]. The reduction of dietary saturated fatty acids (SFAs) primarily lowers large LDL particles, less strongly associated with cardiovascular disease (CVD), while small atherogenic LDL particles, more strongly associated with CVD through their plasma residence time and enhanced oxidative susceptibility, are minimally affected by SFA content in the diet [118]. As protein utilization decreases over the age of 60, protein requirements increase with age (0.9–1.1 g/kg). This can be met by eating meat, especially poultry, fish, milk and dairy products. Approximately 55–60% of the energy should be covered by carbohydrates, preferably from sources such as whole grain products, vegetables, fruits and brown rice. Daily consumption of fruits and vegetables should exceed 400 g, which is also essential to reach the recommended daily intake of 30–40 g of fibers (Table 1). The intake of fats should not exceed 25–30% of the total daily energy intake [119], and vegetable oils should be favored as opposed to animal fats. Data reveal that a diet including vegetable fat rather than animal fat might be beneficial in type 2 diabetes prevention [119]. The weekly consumption of sea fish should be encouraged as well, as they help prevent cardiovascular diseases and certain mental conditions due to their high omega-3 fatty acid content [120,121].

Minerals and vitamins in our diet may also be essential in healthy aging. Both men and women should strive to consume 600 international units of vitamin D daily (e.g., from fish, egg yolk or supplements) [122]. Women aged over 50 and men aged over 70 are also recommended to consume 1200 mg calcium per day as a preventive measure for osteoporotic fractures. As people age, the amount of stomach acid decreases, which may reduce vitamin B_12_ intake and consequentially lead to symptoms such as depression and fatigue [119]. Supplements or fortified foods rich in vitamin B_12_ (e.g., orange juice, milk or yogurt) should also be considered to reach the daily recommendation of 2.4 micrograms of vitamin B_12_ [123]. Finally, consumption of zinc supplements is also advised in advanced age, as zinc helps the normal functioning of the immune system and has anti-inflammatory properties as well [124,125]. Recent studies are also evaluating the potential role of combining dietary interventions with repurposed drugs and supplements targeting aging mechanisms to optimize aging trajectories, including rescue of vascular function and prevention of age-related cognitive impairment.

### 3.2. Healthy Hydration for the Elderly

Appropriate fluid intake is necessary to the physiological ageing process [126]. It is known that dehydration is associated with increased hospitalization, morbidity and mortality. The elderly are vulnerable to hypohydration because of the physiological and cognitive changes occurring at advanced age [127,128].

The European Food Safety Authority (EFSA) published recommendations for age-specific fluid intake, and the current recommendation is 2.0 L/day for adult females and 2.5 L/day for adult males [129]. The European Society for Clinical Nutrition and Metabolism (ESPEN) guidelines [130] recommend slightly less daily fluid intake, as it also takes into account the fluid content of foods, hence: 1.6 L/day for females and 2.0 L/day for males [126].

The state of low body water (hypohydration) has deleterious effects on cardiovascular health. Observational studies in the United States have described that long-lasting low water intake increased the risk for adverse cardiovascular events. There is evidence that acute hypohydration induces endothelial dysfunction, increases sympathetic nervous system activity, and may worsen orthostatic tolerance. Thereby, hypohydration impairs vascular function and blood pressure regulation. This mechanism also plays a role in decreased mental and physical performance [128,131].

Hypohydration can lead to many diseases and health issues, e.g., fractures as a result of falls, urinary infections, constipation, pressure ulcers, kidney problems such as stones and functional impairment. Dehydration is also a risk for acute coronary events (OR 1.16, 95% CI 1.03–1.32), pneumonia (OR 1.23, 95% CI 1.13–1.34) and thromboembolism (OR 1.28, 95% CI 1.14–1.42) [132].

Appropriate fluid intake is important for preserved cognitive function [133]. The abnormalities of water homeostasis can lead to early expressions of neuronal dysfunction, chronic cerebral vasculopathy, brain atrophy and neurodegenerative disease [134]. Water consumption can also positively influence cognitive abilities and mood states [135,136,137].

Adequate hydration is also essential for prevention of chronic diseases (such as diabetes, heart disease, kidney stones or renal failure) or may delay their development [138]. Hypohydration seems to be the initial “cause” of many diseases: diabetes, kidney disease, obesity and cancer, leading to a shortened lifespan. Therefore, proper fluid intake might prevent older people from suffering acute health problems, improve health and reduce the risk of chronic diseases commonly occurring in the elderly [133].

In addition, proper hydration was a contributing factor to a faster, better recovery in a retrospective descriptive study conducted in patients after stroke in an intensive care unit [138].

The increase in the amount of consumed water together with the application of angiotensin antagonists can also reduce body weight and can improve metabolic status [133]. Increased fluid intake is recommended for type 2 diabetic patients as well, which can help prevent and can significantly reduce the occurrence of chronic diseases [133].

### 3.3. Experimental Diets Targeting Healthy Aging

Several types of diets have been developed to promote healthy aging [27]. Intermittent fasting significantly improved several aspects of the quality of life, decreased fatigue and significantly lowered IGF-1, which can act as an accelerator of tumor development and progression [27]. Longevity-promoting diets include caloric restriction, intermittent fasting (ideally with an 11–12 h daily eating period), restriction of methionine intake, and consumption of fruits and vegetables, legumes, whole grains and oilseeds rich in beneficial fats (e.g., nuts) [139]. The emphasis is on the consumption of fresh ingredients rather than semi-prepared or refined foods [139]. The diet is mostly plant-based with 45–60% of the daily calorie requirement provided by unrefined, complex carbohydrates, 10–15% by predominantly plant-based proteins and 25–35% by mostly plant-based fats [139]. The longevity diet combines the beneficial effects of an anti-inflammatory diet rich in vegetables and fruits with the positive aspects of fasting.

Franceschi et al. recently developed a new dietary recommendation for healthy aging, which is based on the Mediterranean diet [140] supplemented with vitamin D [141,142]. This diet does not only decrease the occurrence of age-related diseases, such as cardiovascular disease, cancer and osteoporosis [143], but may also be effective in reducing low-grade systemic inflammation [141,142]. The anti-aging effect of the diet is mediated by other factors apart from inflammation, such as its ability to prevent the shortening of chromosome telomeres and its ability to reduce lipid peroxide, hydrogen peroxide and tumor necrosis factor levels and increase nitric oxide levels [144].

Other dietary approaches for healthy aging have been proposed as well. Several studies have shown, for instance, that restricting calorie intake [101,145,146,147,148,149,150,151,152,153,154,155,156] is an effective way to combat many chronic diseases and metabolic disorders [155,157,158,159]. Other studies have shown that restriction from dietary methionine [160] can delay age-related pathophysiological manifestations. Fasting can mean abstaining from certain foods, for example, meat, or abstaining from foods altogether for a shorter or longer period of time [161] and has been shown to exert multiple beneficial effects in aging [162,163]. Fasting for 12 to 24 h triggers autophagy in cells, which does not only provide energy but also helps cells to renew and survive, and thus slows down the aging process [139]. Fasting methods may activate autophagy, especially during the late portion of the fasting period, and increase the levels of stem cells and regeneration in various tissues, especially during the re-feeding period [139]. Downstream consequences of these changes are improved metabolic function, reduced inflammation with delayed immunosenescence, reduced oxidative damage and improved proteostasis [139,164,165,166]. Because of this, fasting is being investigated as a potential augmentative therapy during cancer therapies [167]. Autophagy impairment has been linked to several diseases, such as Alzheimer’s and Parkinson’s diseases, diabetes, cancer, chronic inflammatory diseases, depression and chronic fatigue syndrome [168,169]. Increased enthusiasm about fasting and its associated benefits [170] has prompted interest from clinical researchers. A recent study found that short-term, time-restricted feeding is safe and feasible in non-obese healthy midlife and older adults [171].

## 4. Nutrition, Diets and Prevention of Cardiovascular Diseases

Unhealthy nutrition plays a central part in the development of cardiovascular diseases [172,173,174,175]. Unhealthy dietary patterns (e.g., low intake of vegetables and fruits, excessive intake of sodium, added sugars, etc.) may lead to conditions such as obesity, diabetes, hypertension and dyslipidemia that greatly burden the cardiovascular system [176,177]. A healthy diet can reduce the risk of cardiovascular disease by up to 30% [176,178]. The Mediterranean diet, as seen in PREDIMED (Prevención con Dieta Mediterránea) and PREDIMED-Plus studies, may have a positive effect on cardiovascular disease development [178,179]. This effect may be partially mediated by their weight-reducing effect [178,179]. By normalizing body weight, the unwanted consequences of obesity and overweight can be prevented (see Table 2). The cardioprotective elements of a balanced diet are summarized in Table 3, while the various nutritional recommendations to decrease cardiovascular risk are displayed in Table 4 [180,181,182].

In patients affected by cardiovascular diseases, the total fat intake needs to be reduced, and saturated fatty acids should be replaced with unsaturated fatty acids [183]. The risk of cardiovascular diseases can be reduced by 2–3% by replacing one energy percent of saturated fatty acid with polyunsaturated fatty acids. The intake of omega-3 fatty acids, including eicosapentaenoic acid and docosahexaenoic acid, are also highly recommended because of their cardioprotective properties. For the omega-3, the minimum healthy target intake is a combined 500 mg/day of eicosapentaenoic acid (EPA) and docosahexaenoic acid (DHA), and moderate intake (around 10g/day) of linoleic acid is recommended [184]. There are no universal guidelines for the recommended ratio of omega-6 to omega-3 fatty acids for the prevention of cardiovascular diseases, but some reports suggest that a proportion of 1:1 to 4:1 (omega-6: omega-3) is ideal [185]. The consumption of oil seeds and fish is also crucial for prevention, as 30 g of oilseeds daily decreases the risk of cardiovascular diseases by about 30% while eating fish twice a week reduces the risk of stroke by 6%. Conversely, the intake of trans fatty acids should be kept below 1 energy%, as trans fatty acids increase low-density lipoprotein levels and lower high-density lipoprotein levels, which are very strong risk factors for cardiovascular diseases [180,181,182,186]. Two recent studies concluded that there is inconsistent evidence on the relation of fatty acids to coronary heart disease and stroke risk, and that higher intakes of total and saturated fats were associated with lower likelihood of having hypertension, while higher intakes of short-chain saturated fatty acids (SCFAs) were inversely associated with dyslipidemia and diabetes [187,188]. SCFA metabolic remodeling was related to cognitive benefits, better antioxidant capacity, the attenuation of inflammation and longevity [189].

Sodium and potassium are also important dietary elements of cardiovascular disease prevention. Cutting down on salt intake by 1 g/day reduces blood pressure by 3.1 mmHg in patients with hypertension [190]. In healthy individuals, the decrease is less pronounced albeit still significant with an observed decrease of 1.6 mmHg [191]. In contrast, potassium intake should be encouraged as it has been described to have a positive effect on blood pressure (at least 4700 mg of potassium/day) [172,173].

Fiber may also play an important role in cardiovascular disease prevention. A 7 g/day increase in fiber intake reduces cardiovascular disease risk by 9%, while a consumption of 10 g/day fiber also lowers the risk of stroke by 16% and the risk of type 2 diabetes mellitus by 6%. A part of the beneficial effects of fiber could be mediated by their body weight lowering effects [192]. Several human experiments support that increased dietary fiber intake is accompanied by weight loss, prolonged mealtimes, increased satiety, improved glucose homeostasis and fatty acid metabolism, decreased calorie intake and increased microbiome diversity [193,194]. Fiber also mitigates the deposition of cholesterol [195] and lipoproteins [196,197] in the vessels by reducing the adsorption of bile acids as well as by modifying the microbiome [198].

### Nutrition, Diets and Endothelial Protection

It is well-established that endothelium-derived nitric oxide (NO) is an important vasodilator gasotransmitter that regulates smooth muscle mediated vascular resistance and thereby organ blood flow. In addition to maintaining normal tissue perfusion, endothelium-derived NO is also involved in numerous vasoprotective signaling pathways. For instance, it inhibits platelet aggregation and inflammatory cell adhesion to endothelial cells. NO also regulates cell division and survival, disrupts pro-inflammatory cytokine-induced signaling pathways, preserves progenitor cell function, and regulates mitochondrial function and cellular energy metabolism.

Cocoa flavanols present in dark chocolate, soy isoflavones, beetroot juice, garlic, organ meat, leafy greens, citrus, pomegranate, nuts and seeds, and watermelon have been implicated in improvement of NO-mediated vasodilation [199]. By drinking beetroot juice, the high content of nitrates (NO_3_) is partially absorbed into blood through intestinal mucosa and converted to nitrite (NO_2_) through a non-enzymatic process. Recycled NO_2_ is reabsorbed and concentrated by the salivary glands and then secreted into saliva to be converted to NO by symbiotic bacteria located both in the oral cavity and stomach [200]. Due to the high dietary NO_3_ content, beet consumption is associated with numerous benefits, including improved cognitive function, enhanced athletic performance and lower blood pressure [201,202,203]. Garlic (*Allium sativum*) and similar plant species are rich sources of sulfur compounds. The sulfur constituents of garlic have been involved in the regulation of vascular homeostasis and the control of metabolic systems linked to nutrient metabolism. Recent evidence implicated one of these sulfur compounds, diallyl trisulfide (DATS), to alter the levels of gaseous signaling transmission of NO in mammalian tissues [204]. Leafy green vegetables (spinach, arugula, kale) and cruciferous vegetables are also very rich in NO_3_. The results of a clinical meta-analysis indicated a significant 15.8% reduced incidence of cardiovascular disease and slowed the progression of cognitive decline with the intake of green leafy vegetables [205]. Despite the fact that specific nutrients responsible for these effects are still under investigation, some of the hypothesized intervening variables found in leafy greens include folic acid, the antioxidants beta-carotene and vitamin E, soluble fiber, calcium and vitamin K [206].

Citrus fruits such as tangerines, oranges, lemons, grapefruits and kumquats are very rich in vitamin C, or ascorbate, which in preclinical studies has been shown to enhance endothelial nitric oxide synthase (eNOS) activity by stabilizing the eNOS cofactor tetrahydrobiopterin (BH4) [207]. In another study, vitamin C improved microvascular reactivity and peripheral tissue perfusion in septic shock patients [208]. More evidence from studies in patients with chronic endothelial dysfunction due to atherosclerosis, hypertension or diabetes have shown the beneficial effects of vitamin C on endothelial and nitric oxide dependent vasodilation. The rapid measured effects of ascorbate (only 1-h post-injection) may be mediated by increased NO availability, either through enhanced synthesis mediated by BH4 recycling, direct reduction of nitrite to NO, release of NO from nitrosothiols or by scavenging superoxide (O_2_^−^) that would otherwise react with NO to form peroxynitrite (ONOO^−^).

Pomegranate (*Punica granatum* L.) is another food that is both rich in NO_3_ and vitamins including ascorbate. Clinical investigation of the acute effects of pomegranate juice has attributed its vasculoprotective effects to the presence of hydrolysable tannins ellagitannins, ellagic acid, anthocyanins, and flavonoids have demonstrated in vivo that they reduce oxidative stress and platelet aggregation, diminish lipid uptake by macrophages, positively influence endothelial cell function and are involved in blood pressure regulation [209,210].

Berries are high in anthocyanin flavonoids, a polyphenol subtype with vascular beneficial properties including increased NO production and reducing oxidative stress and inflammation [211]. It has been previously shown that polyphenols from wine and grape berries’ extracts exert robust vasodilator effects, due to the increased eNOS expression and activity, consequently increasing NO production [212]. Moderate consumption of wine is recommended due to the deleterious effects of excessive alcohol consumption, which exerts pro-oxidant effects and depletes NO bioavailability via the endogenous NOS inhibitor asymmetric dimethylarginine, which mediates the decreased synthesis of NO [213].

Nuts and seeds are rich in L-arginine, the amino acid precursor for the production of NO. Watermelon contains L-citrulline, and recent evidence suggests that in cerebral arteries, L-citrulline is converted to L-arginine most likely via an argininosuccinate pathway to synthesize NO [214]. Multiple other studies have associated L-citrulline supplementation with increased NO synthesis, decreased blood pressure and increased peripheral blood flow [215].

Similarly, other dietary elements have been associated with decreased endothelium-deprive NO production, effectively worsening cerebrovascular function and cognitive outcomes. High table salt (NaCl) intake has been shown to reduce NO drastically; studies showed that it is recommended not to exceed 5 g/day, while the World Health Organization suggests consuming no more than 1 teaspoon of salt per day. Similarly, elevated blood glucose levels destabilize NO bioavailability, and recent studies suggest controlling glucose levels by decreasing consumption of refined sugars. High consumption of saturated fats increases low-density lipoprotein (LDL) cholesterol. Clinical studies have revealed that participants undergoing an LDL-lowering diet benefitted from improved endothelial function when compared with placebo participants. Interestingly, most of the deleterious nutrients are readily found in the standard American diet (SAD), which could in part explain why individuals in North America have higher rates of vascular inflammation, heart disease, brain strokes, coronary artery disease, and cognitive impairment in addition to being more likely to experience weight gain and constipation compared to other countries. In contrast, adherence to the Mediterranean diet was inversely associated with the prevalence of metabolic syndrome in the elderly [216]. Walnuts, part of the Mediterranean diet, significantly decreased triglyceride, total cholesterol and LDL cholesterol concentrations and presented no consequences on anthropometric and glycemic parameters [217]. Hydrosoluble micronutrients (such as polyphenols), lipophilic compounds (tocopherols and tocotrienols; n-3 PUFAs and n-6 PUFAs) and other plant molecules exert their antioxidant action through multiple mechanisms, including the activation of the Nrf2/ARE pathway and down-regulation of the NF-kB pathway, directly implicated in the inflammatory response [217].

The most consistently reconfirmed biological adaptations to the Mediterranean diet include strong antioxidant protection against oxidative stress and inflammation, a lipid-lowering effect, modulation of humoral factors involved in the pathogenesis of cancer, inhibition of energy-sensing pathways and better gut microbiota-mediated release of metabolites beneficial to metabolic health. Adequate nutrition is therefore a crucial component to maintain healthy cerebrovasculature to lead a healthy lifestyle, leading to improved cognitive outcomes in aging.

## 5. Nutrition, Diets and Healthy Brain Aging

### 5.1. Prevention of Cognitive Decline and Dementia

A healthy diet can also help prevent cognitive decline and maintain cognitive functioning in older individuals [218]. The occurrence of dementia increases with age, doubling every 5 years after the age of 65 [219]. Risk factors of dementia can be divided into two groups: modifiable and non-modifiable factors (shown in Table 5). Non-modifiable factors include age, sex and genetic factors, while modifiable factors include vascular and metabolic (hypertension, cholesterol, homocysteine, atherosclerosis, diabetes mellitus, etc.), lifestyle (diet, physical and mental activity, smoking, alcohol consumption, obesity, etc.), environmental, (trauma, pesticides, etc.), and disease-related factors (depression). Controlling modifiable risk factors can reduce the incidence of dementia by up to 35% [220]. Studies focusing specifically on the prevention of cognitive decline, such as the Finnish Geriatric Intervention Study to Prevent Cognitive Impairment and Disability Study (FINGER Study), also support these observations [221].

Modulation of metabolic pathways is one of the most promising strategies to promote resilience and delay aging [222]. Diet is one of the most important lifestyle factors associated with dementia [223], including Alzheimer’s disease [224]. The Mediterranean diet has a risk-reducing effect for dementia, which may be mediated by its cardioprotective effect [225]. The MIND diet (Mediterranean-DASH diet Intervention for Neurodegenerative Delay), which is based on the Mediterranean diet (see Table 6), has also been associated with better cognitive functioning [226,227,228].

The role of vitamin D and the vitamin B complex (mainly vitamin B_12_ and B_3_ and folic acid) has been confirmed as a protective factor for dementia [229,230,231]. While vitamin D is a potential anti-inflammatory and antioxidant vitamin, vitamin B may enhance the production of certain neurotransmitters and exert an antioxidant effect. The antioxidant properties of vitamin B_12_ are accomplished by different mechanisms, including the direct scavenging of reactive oxygen radicals, particularly superoxide in the cytosol and mitochondria, and indirectly by stimulating the scavenging of reactive oxygen radicals through the preservation of glutathione.

Mitochondria play a crucial role in mediating and amplifying the oxidative stress that contributes to the process of aging [232,233,234,235]. A growing number of studies have focused on elucidating the role of mitochondrial dysfunction [236], mitochondria-derived mitokines [237] and peptides [238] in the pathogenesis of age-related cognitive impairment. It has become very clear that significant pathophysiological disturbances are associated with mitochondrial aberrations in clinical studies [239] and preclinical models of aging. Mitochondria DNA copy number was found to be a predictor of physical performance in older men [240]. Possibly, the most important lifestyle factor determining adequate mitochondrial function is appropriate nutrition.

Important dietary compounds such as curcumin, astaxanthin, resveratrol, hydroxytyrosol, oleuropein, and spermidine (which are present in the Mediterranean diet) have been shown to exert powerful beneficial effects, enhancing mitophagy (the degradation of damaged mitochondria) by upregulating mitophagy mediators, and promoting new mitochondria biogenesis [241]. Due to this evidence, many laboratories interested in developing anti-aging interventions are targeting mitochondrial dysfunction [242,243,244], and investigating novel roles of mitochondria in longevity.

Caffeine has also sometimes been described to play a protective role in the development of certain forms of dementia, such as Alzheimer’s disease [245]. Caffeine is a non-selective antagonist of adenosine receptors, demonstrating a structural similarity to adenosine. The blocking of these receptors (A1, A2) at lower concentrations of caffeine (plasma concentration <250 µM) modulates glutamatergic, cholinergic, dopaminergic, serotoninergic and noradrenergic neurotransmission. Moreover, caffeine is an agonist of ryanodine receptors as well that play an important part in calcium release from the endoplasmic reticulum. Caffeine is also a non-selective competitive inhibitor of phosphodiesterase, the enzyme degrading cyclic adenosine monophosphate. However, these latter effects are only possible at higher doses (plasma concentration >250 µM). Moreover, caffeine also interferes with γ-aminobutyric acid type A receptors, protects against cell damage by reducing oxidative stress and may exert anti-inflammatory activity by decreasing proinflammatory (C-reactive protein, IL-1β, IL-6, IL-18, TNF-α) and increasing anti-inflammatory (IL-10, adiponectin) marker levels [245]. According to the European Food Safety Authority, 400 mg (approximately 5.7 mg/kg bodyweight) of caffeine a day does not raise any safety concerns (except in pregnancy) [246].

More than a dozen epidemiological studies have reported that reduced intake of omega-3 fatty acids or fish consumption is also associated with increased risk for age-related cognitive decline or dementia, such as Alzheimer’s disease [247,248,249]. Docosahexaenoic acid is neuroprotective via multiple mechanisms that include the reduction of arachidonic acid metabolites and the increase of trophic factors or downstream trophic signal transduction. Docosahexaenoic acid is specifically protective against Alzheimer’s disease via additional mechanisms as well. It limits the production and accumulation of the amyloid beta peptide that is widely believed to drive the disease; and it also suppresses several signal transduction pathways induced by amyloid beta peptides, including two major kinases that phosphorylate the microtubule-associated protein tau and promote neurofibrillary tangle pathology. Results to date suggest that docosahexaenoic acid may be more effective if it is supplemented early or used in conjunction with antioxidants [250].

Recent studies demonstrate that the microbiota is an important modifier of risk factors for age-related neurodegenerative diseases and frailty [251,252,253,254,255,256] and that changes in microbiome-derived metabolites modulate neuroinflammation and age-related cognitive dysfunction [257].

### 5.2. Nutritional Factors Contributing to Cerebrovascular Protection

The human brain is an exquisitely complex organ with high metabolic demands. For the average adult in a resting state, the brain consumes about 20 percent of the body’s energy expenditure despite it only comprising 2 percent of the body’s weight. Unlike muscles, which possess energy reserves, the brain needs to be constantly supplied with oxygen and nutrients to function adequately. This is ensured by tightly controlled regulation of cerebral blood flow (CBF). Changes in neuronal activity require prompt CBF adjustments to maintain cellular homeostasis [258,259], which is accomplished via neurovascular coupling (NVC; also known as functional hyperemia) [260,261]. Recent studies demonstrate that aging is associated with diminished NVC responses [262] and that impaired NVC is causally related to cognitive decline [263,264].

Preclinical and clinical evidence shows that the endothelium plays a critical role in mediating NVC responses [265,266,267,268]. Pathological conditions that promote endothelial dysfunction (including obesity, diabetes mellitus, hypertension, among others) [269] were shown to exacerbate age-related endothelial dysfunction and neurovascular impairment [270], contributing to cognitive decline. Cerebrovascular endothelium is also important to maintain the integrity of the blood–brain barrier (BBB). Disruption of the BBB also has been causally linked to cognitive impairment [271,272].

Cerebrovascular and endothelial health are importantly modulated by nutritional factors and dietary patterns [100,109,154,273,274]. High fat diets and diets containing high levels of methionine were shown to exert deleterious effects on endothelial cells, promoting vasodilator dysfunction, BBB disruption and neuroinflammation, microvascular rarefaction and neurovascular uncoupling [100,275,276]. Pro-inflammatory diets such as those high in processed foods and refined sugars also have a deleterious effect on cerebral blood vessels and brain function [277]. Additionally, many elderly individuals live with an array of systemic vascular comorbidities that exacerbate the diverse pathophysiological processes promoting cognitive dysfunction [278,279].

Diets enriched for polyphenols (e.g., resveratrol) confer multifaceted anti-aging vasoprotective effects [274,276,280]. There are aso studies showing that dietary intake of cocoa flavanols protects humans against vascular disease, as evidenced by improvements in peripheral endothelial function [281]. An interesting study tested the effects of cocoa consumption in 60 participants in which 17 volunteers presented with impaired NVC [282]. Participants with intact NVC responses showed no significant benefits from cocoa consumption. However, subjects with impaired NVC experienced a dramatic doubling in NVC responses after just a month of cocoa supplementation, while scores on standard cognitive tests increased by 30 percent [282].

Aging is associated with NAD^+^ depletion, a change that is proposed to be a major contributor to organ-specific aging processes [283,284]. NAD^+^ is a co-substrate for sirtuin enyzmes, which are regulators of important cellular processes of aging, mitochondrial function, stress resilience apoptosis and inflammation in the vasculature [285,286]. Emerging evidence shows that vascular and cerebrovascular aging is driven by NAD^+^ depletion [81,82]. Recent studies show that the restoration of NAD^+^ levels via administration of NAD^+^ precursor nicotinamide mononucleotide (NMN) in aged murine models leads to the rescue of cerebrovascular endothelial function and NVC [81,82]. Clinical trials utilized a different NAD^+^ precursor, nicotinamide riboside (NR), which successfully increased NAD^+^ levels is humans subjects [287,288], making it a possible candidate to restore NAD^+^ concentration in order to achieve anti-aging effects.

### 5.3. Prevention of Depression and Other Geriatric Psychological Disorders

Studies indicate that our diet may influence the pathogenesis of psychological disorders, such as depression [289,290,291,292,293]. A recent meta-analysis [289] showed that [5,19,33,294,295,296,297,298] a Mediterranean diet may reduce the risk for depression by up to 30% [289,290,291]. In contrast, consumption of a Western diet with its high intake of saturated fats, refined carbohydrates and processed foods, is associated with poorer mental health indicators [292]. Similar observations were made in the SMILES study [299], in which one group of participants was prescribed a special diet for 12 weeks, while the control group received only peer support during the same time. The 12-week program consisted of a diet made up of the following elements: whole grains (5–8 times per day); vegetables (6 times per day); fruits (3 times per day); leguminous vegetables (3–4 times per week); low-fat, sugar-free dairy products (2–3 times per day); raw and unsalted grains (once per day); fish (twice a week); lean red meats (3–4 times a week); chicken (2–3 times a week); eggs (no more than 6 times a week); olive oil (3 tablespoons a day); and a limited amount of extra sugary soft drinks, processed meats and fast foods (up to 3 times a week). During follow-up, participants in the 12-week diet program reported significantly fewer depressive symptoms than the control group. Similar conclusions were reached by other studies as well [300].

Fruits and vegetables seem to be especially important in the prevention of psychological disorders. People who eat more fruits and vegetables have fewer mental disorders including depression, stress and negative mood [301]. They also report higher rates of happiness, good mood and life satisfaction [302]. Moreover, the association between fruit and vegetable intake and mental health appears to be dose-dependent: the higher the intake, the better the mental health was reported by study participants [303,304]. The consumption of fruits and vegetables exerts its positive effect not only through its vitamin, antioxidant and fiber content, but also by affecting the gut microbiome [305].

Another frequently examined group of nutrients in terms of mental illnesses is fatty acids. Polyunsaturated fatty acids make up about half of the grey matter, of which one-third is made up of fatty acids from the omega-3 family [306]. As omega-3 is an essential fatty acid, it cannot be produced by the body and must be obtained through diet [307]. Results from epidemiological and neurobiological studies have suggested that a relative deficiency of omega-3 polyunsaturated fatty acids may predispose the individual to psychological disorders, including depression. Several other studies have also highlighted the association between lower omega-3 unsaturated fatty acid levels and depression [308,309]. Furthermore, several preliminary small case-control clinical trials also indicate that omega-3 fatty acid supplements may be useful in treating symptoms of depression after previous antidepressant treatment has failed [307,310]. The risk-reducing effect of omega-3 fatty acids may be conveyed by their anti-inflammatory properties [307,308,309,310].

Vitamins may also have an impact on our mental health. It has been long known that higher levels of homocysteine and lower levels of vitamin B_9_ (folic acid) increase the risk of depression in older ages. A randomized, double-blind, placebo-controlled clinical trial also showed that vitamin B_6_, B_9_ and B_12_ supplementation, along with lower homocysteine levels, enhances the effect of standard antidepressant treatment in old age [311]. Finally, both vitamin D [312] and vitamin B complex (B1/B6/B12: 180/180/1 mg/body weight kg, daily) [311] have been described to improve stress tolerance, anxiety and depressive symptoms as well. The association appears to be driven by the homeostatic, trophic and immunomodulatory effects of vitamin D. Furthermore, vitamin D also modulates the hypothalamic–pituitary–adrenal axis, which regulates the production of the monoamine neurotransmitters epinephrine, norepinephrine and dopamine in the adrenal cortex and also protects against the depletion of dopamine and serotonin [313]. The vitamin B complex, on the other hand, may exert its positive effect on mental health by improving the carbohydrate metabolism in nerve cells [311].

## 6. Conclusions

Nutrition and diets play a central part in the development of several age-related diseases, such as cardiovascular disease, neurodegenerative disease and dementia. A well-managed nutrition and a diet adapted to age is essential to maintain mental freshness and a good quality of life at old age. Diets also influence the process of aging itself. By adhering to certain diets, such as the Mediterranean diet, the onset of age-related diseases can be prevented or delayed. This is achieved directly by certain elements included in the diet, for instance, fruits, vegetables and omega-3 fatty acids, and indirectly by their positive effect on bodyweight management. Moreover, the introduction of fasting episodes into our diet may also contribute to healthier aging. Diet recommendations change with age, and this must be taken into consideration when developing a diet tailored to the needs of elderly patients. Future and ongoing clinical studies (e.g., ClinicalTrials.gov Identifier: NCT03702335 (Impact of Comprehensive Dietary Counseling on Dietary Quality, Mental Health, and Quality of Life in Older Adults); NCT05593939 (Slow Age: Interventions to Slow Aging in Humans); NCT02751866 (Early Intervention in Cognitive Aging) on complex anti-aging dietary interventions translating the results of promising preclinical investigations are expected to lead to novel nutritional guidelines for older adults in the near future.

## Figures and Tables

**Table 1 nutrients-15-00047-t001:** Summary of the health benefits of dietary fiber.

Effect	Health Benefit
Metabolic	Improved insulin sensitivity, reduced risk of developing T2D, improved glycaemic status and lipid profiles, reduced body weight and abdominal adiposity
Gut microflora	Gut microbial viability and diversity, metabolites from gut microflora
Cardiovascular	Chronic inflammation, cardiovascular risk, mortality
Depression	Chronic inflammation, gut microbiota
Gastrointestinal localized	Colonic health and integrity, colonic motility, colorectal carcinoma

Source: ref [122].

**Table 2 nutrients-15-00047-t002:** Pathological consequences of obesity.

I. Metabolic consequences	Diabetes Insulin resistance Gout Persistent inflammation	Metabolic syndrome Dyslipidemia Hyperuricemia
II. Cardiovascular diseases	Hypertension Coronary heart disease	Venous thromboembolism Stroke Congestive heart failure
III. Respiratory diseases	Asthma Sleep apnea syndrome (OSAS)	Hypoxia Hypoventilation syndrome
IV. Tumors	Esophageal-, intestinal-, rectal-, liver-, gall bladder-, pancreas-, kidney tumors Leukemia, lymphoma, multiple myeloma In women: endometrial, cervical, ovarian, breast cancer Men: prostate cancer	
VI. Gastrointestinal	Gallbladder diseases Non-alcoholic fatty liver disease Gastroesophageal reflux Abdominal and inguinal hernia	
VII. Genitourinary system and reproductive organs	Urine loss Irregular menstruation Hirsutism Hypertonia Miscarriage Esophageal abnormalities Birth defects	Gestational diabetes Infertility Polycystic ovary Pre-eclampsia Large fetus Fetal distress Caesarean section
VIII. Psychological and social disorders	Low self-esteem Stigmatization	Anxiety, depression Work and employment problems
IX. Other pathologies	Nephrosis syndrome Complications of anesthesia Idiopathic intracranial hypertension	Lymphoedema Proteinuria Periodontal diseases Skin infections

Source: ref [175].

**Table 3 nutrients-15-00047-t003:** Cardioprotective factors in nutrition based on the 2016 ESC guideline.

Saturated fatty acid <10 E%, achieved by replacing the excess with polyunsaturated fatty acid in the diet.
The amount of trans fatty acid should be reduced as much as possible by limiting the consumption of processed products and keeping the intake of natural trans fatty acid below <1 E%.
<5 g/day salt intake.
30–45 g/day dietary fiber, preferably whole grains.
≥200 g fruit (2–3 portions/day)
≥200 g vegetables (2–3 portions/day)
Fish 1–2 times/week, of which one should be fatty fish (high in fat).
30 g/day unsalted oilseeds.
Limit on alcoholic drinks: 2 glasses/day (20 g/day of alcohol) for men, 1 glass/day (10 g alcohol/day) for women.
Avoid sugary and alcoholic drinks.

Source: ref [180].

**Table 4 nutrients-15-00047-t004:** Comparison of different recommendations to reduce cardiovascular risk.

	2016 European Guideline	2014 NICE Guideline	2020 NICE Pathway
Vegetables	≥2 dose/day	2–3 dose/day	2–3 dose/day
Fruit	≥2 dose/day	2–3 dose/day	2–3 dose/day
Fish	≥2 dose/week	≥2 dose/day	≥2 dose/day
Fat (saturated)	<10 E%	<30 E%	30–35 E%
Added fats		olive/rape and products made from these fats and oils	olive/rape and products made from these fats and oils
Fiber (g/day)	30–45		
Sodium (mg/day)	<2500		
Oil seeds		4–5 dose/week	4–5 dose/week
Pulses vegetables		4–5 dose/week	4–5 dose/week

Source: ref [180,181].

**Table 5 nutrients-15-00047-t005:** Risk and protective factors for dementia.

Risk Factors	Protective Factors
▪ Age	▪ Genetic factors: some mutations in the Amyloid Precursor Protein gene, APOE ε2 allele
▪ Genetic factors: familial predisposition, APOE ε4 allele, other genes	▪ Lifestyle factors: education, intellectual work, extensive social contacts, mental stimulation, physical activity
▪ Vascular and metabolic factors: arteriosclerosis, stroke, diabetes mellitus, hypertension, obesity, high cholesterol in middle age	▪ Diet: low alcohol, Mediterranean diet, unsaturated fats, oils, vitamin B_12_, folic acid, vitamin D
▪ Lifestyle factors: smoking, inactivity, heavy alcohol consumption	▪ Medications: antihypertensives, statins, hormone replacement therapy, NSAIDs
▪ Diet: saturated fats, hyperhomocysteinemia, vitamin deficiency	
▪ Other factors: depression, trauma, toxic effects, infectious diseases	

**Table 6 nutrients-15-00047-t006:** The MIND (Mediterranean-DASH Diet Intervention for Neurodegenerative Delay) diet focuses on healthy foods.

Include These	Limit These
▪ Green leafy vegetables: every day	▪ Red meats
▪ Other vegetables: at least once per day	▪ Butter and stick margarine: less than 1 tablespoon per day
▪ Nuts: every day	▪ Cheese: less than one serving per week
▪ Berries: at least twice per week	▪ Pastries and sweets: limit
▪ Beans: every other day	▪ Fried or fast food: less than one serving per week
▪ Whole grains: three times per day	
▪ Fish: at least once per week	
▪ Poultry: at least twice per week	
▪ Olive oil	
▪ Wine: one glass per day	

Source: ref [228].

## Data Availability

Not applicable.

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
