# Peer review of "Nutrition Strategies Promoting Healthy Aging: From Improvement of Cardiovascular and Brain Health to Prevention of Age-Associated Diseases"

_nutrients, 2022, doi:10.3390/nu15010047_

Round 1

Reviewer 1 Report (Previous Reviewer 2)

Thank you for asking me to review this revised version of the manuscript.
I feel it has now strongly improved, and I think it is worth publication in this journal.
The topic is of interest, well presented and described, and it offers a wide sight of nutritional considerations for healthy aging. I have no further comments.

Author Response

Thank You for your valuable comments, Thank You very much!

Reviewer 2 Report (Previous Reviewer 1)

The authors addressed some questions and suggestions. However, to one of my previous suggestions, the authors responded “... many of references citing in vivo studies were not removed.” I agree that “Citing in vivo studies allows us further insight into mechanisms where it is difficult to examine in human subjects” but, based on what you mentioned in Abstract “A literature review was conducted to survey the recent ... findings...” and the fact that this field is very dynamic, I still recommend replacing the old references with newer ones. Please see the followings: 

Lines 75-76: “Part of these age-related changes are driven by increased production of oxygen free radicals[75-98]” - the references 75 to 87 are old (from 2002 to 2014) and should be replaced; in PubMed, for “oxygen free radicals”, there are 41,532 results for the last 5 years!

Line 89: the sentence says “methionine diets[125-133]” - the references 127, 129-133 are old (1999-2003) and should be replaced; in PubMed, for “methionine diets”, there are 1,757 results for the last 5 years!

Line 218: restricting calorie intake[80,84,110,118,168-181]” - the references 80, 84, 110, 118, 168-178 are old (even 1987) and should be replaced; in PubMed, for “restricting calorie intake”, there are 676 results for the last 5 years!

Lines 541-542: “Diets enriched for polyphenols (e.g. resveratrol) confer multifaceted anti-aging vasoprotective effects[85,89,177,286,303,307,312-316].” - many references are older than 5 years and should be replaced; in PubMed, for “polyphenols” and “resveratrol”, there are 23,358 and 6,308 results, respectively, for the last 5 years!

Author Response

Our response: We thank reviewer for the kind words, and the suggestion about replacing further references. We understand that the listed statements could be supported by newer publications, and in order the address this issue, we have replaced several references, and in other cases, where we found that the already listed references supported our statment, we have only deleted the references that were citing publications published before 2017.

Reviewer 3 Report (New Reviewer)

I read with great interest and am very satisfied with the quality of the manuscript, which is in line with the parameters, and quality standards of the journal.

This literature review aims to describe the role of nutrition and macronutrients in the prevention of various pathologies, especially of a cardiovascular nature. While the manuscript is well structured and comprehensive, I believe the authors should enhance the discussion by including a chapter on COVID-19 and nutrition. It is known, in fact, that this pathology represents one of the major global problems, which mainly affects the cardio-pulmonary system and is more fatal in the elderly. The insertion of this new paragraph would ensure a more complete view of the topic and would be much more appreciated by readers. I recommend a review, which may be useful to the authors (DOI: 10.3390/ijms23169136).

Author Response

Our response: We thank reviewer for the kind words and for the suggestion about adding a chapter on COVID-19 and nutrition. Paralell with the peer reviews, we received an academic editorial request to narrow the focus of our review, hence, several sections were removed (osteoporosis, cancer). To thoroughly address the suggested topic, apart from Covid-19, a chapter about communicable diseases and diseases of the cardiopulmonary system should have been included as well. Currently, the manuscript focuses mainly on nutritional aspects of cardiovascular disease, cerebrovascular disease, and healthy brain aging. Therefore, the topics mentioned above became out of the scope of current review.

This manuscript is a resubmission of an earlier submission. The following is a list of the peer review reports and author responses from that submission.

Round 1

Reviewer 1 Report

This review presents a nutrition strategy for healthy aging. The subject is very important, the authors have extensive background in public health and health promotion, however, the manuscript needs improvements to be considered for publishing. Next are my suggestions:

- authors should use a new title, as in the manuscript cerebrovascular and brain health is part of the research, besides prevention diets of CV diseases, osteoporosis, tumors, eye-related diseases

- author may ensure that they have adhered to the journal's referencing style throughout 

- the description of the methods used should be improved

- the structure of the document could be changed - section 3 and section 10 should be combined 

- there are 394 references, but many of them are old and very old

- there are too many in vivo studies cited even though, in Conclusion, the diet recommendations are for elderly

- please update the whole manuscript with new citations, not older than 5 years. There are numerous new titles in this field, many of them published recently in Nutrients; following are just a few new titles that I recommend: 

New evidence shows that the response to dietary fat intake may be based on individual circumstances and the rise in LDL cholesterol caused by saturated fats may represent a normal rather than a pathologic response, with different factors, such as gut microbiotamediating the response (doi: 10.1093/ajcn/nqaa322).

The gut microbiota alteration was correlated with human neurodegenerative and brain-related diseases including Alzheimer’s and Parkinson’s (doi: 10.3390/nu14193967).

Hydrosoluble micronutrients (such as polyphenols), lipophilic compounds (tocopherols and tocotrienols; n-3 PUFAs and n-6 PUFAs), and other plant molecules exert their antioxidant action through multiple mechanisms, including the activation of the Nrf2/ARE pathway and down-regulation of the NF-kB pathwaydirectly implicated in the inflammatory response (doi: 10.3390/antiox11071412).

The reduction of dietary SFAs primarily lowers large LDL particles, less strongly associated with CVD, while small atherogenic LDL particles, more strongly associated with CVD through their plasma residence time and enhanced oxidative susceptibility, are minimally affected by SFA content in the diet (doi: 10.1093/ajcn/nqaa111).

Two recent studies concluded that there is inconsistent evidence on the relation of fatty acids with coronary heart disease and stroke risk ( doi: 10.1161/JAHA.119.013131) and that higher intakes of total and saturated fats were associated with lower likelihood of having hypertension, while higher intakes of short-chain saturated fatty acids (SCFAs) were inversely associated with dyslipidemia and diabetes (doi: 10.3390/nu14204294).

SCFA metabolic remodeling was related to cognitive benefits, better antioxidant capacity, the attenuation of inflammation, and longevity (doi: 10.3390/nu14204420).

Data reveal that a diet including vegetable fat rather than animal fat might be beneficial in type 2 diabetes prevention (doi: 10.1371/journal.pmed.1003347). 

Most dietary guidelines recommend total fat intakes of 30 to 35% of total energy intake, for MUFA 10-25%, for PUFA 6-11%, for SFA 7-11%, for 200-300 mg/d for dietary cholesterol (doi: 10.1159/000515671)

Adherence to the Mediterranean diet was inversely associated with the prevalence of metabolic syndrome in elderly (doi: 10.1080/09637486.2016.1221900). Walnuts, part of the Mediterranean diet, significantly decreased triglyceride, total cholesterol, and LDL cholesterol concentrations and presented no consequences on anthropometric and glycemic parameters (doi: 10.3390/antiox11071412).

Intermittent fasting significantly improved several aspects of the quality of life, decreased fatigue, and significantly lowered IGF-1, which can act as an accelerator of tumour development and progression (doi: 10.3390/nu14194216).

A clearly distinction should be made between in vivo studies (many cited in this manuscript) and clinical studies, as the results of human studies are not as promising as preclinical reports, in human studies a decrease in total fat intake did not prove to be an effective strategy to combat cancer (doi: 10.3390/ijms21114114).

Author Response

Response to the Comments from the Reviewers

Manuscript No: nutrients-2031422

Title: Aging healthily - Nutritional strategy considering typical dis-eases of the aging population, from prevention of geriatric diseases to promotion of cerebrovascular and brain health

Thank You for your comments and recommendations on our manuscript. We have incorporated the suggestions recommended. We feel that the amendments further improved the scientific quality of the manuscript.

Reviewers' comments:

Reviewer 1

Comments and Suggestions for Authors

This review presents a nutrition strategy for healthy aging. The subject is very important, the authors have extensive background in public health and health promotion, however, the manuscript needs improvements to be considered for publishing. Next are my suggestions:

  1. authors should use a new title, as in the manuscript cerebrovascular and brain health is part of the research, besides prevention diets of CV diseases, osteoporosis, tumors, eye-related diseases

Reply: Thank You very much, the title of the manuscript has been changed.

  1. author may ensure that they have adhered to the journal's referencing style throughout

Reply: Thank You very much for your comments, the references have been corrected in the manuscript.

  1. the description of the methods used should be improved

Reply: Thank You very much for Your comments, the methods has been corrected in the manuscript.

  1. the structure of the document could be changed - section 3 and section 10 should be combined

Reply: Thank You very much for your comment, the structure of the document has been changed in the manuscript.

  1. there are 394 references, but many of them are old and very old

Reply: Thank You very much for your comment, I have changed the references in the manuscript.

  1. there are too many in vivo studies cited even though, in Conclusion, the diet recommendations are for elderly

Reply: Thank You very much, we have improved the text of the conclusion.

  1. please update the whole manuscript with new citations, not older than 5 years. There are numerous new titles in this field, many of them published recently in Nutrients.

Reply: Thank You very much, I have updated the citations.

Thank You very much for the work and help of the Reviewers, we feel that improving the manuscript has greatly improved the scientific quality of the article.

Sincerely:

Monika Fekete MD

Reviewer 2 Report

The narrative review "Nutrition strategy for healthy aging: from prevention of geriatric diseases to promotion of cerebrovascular and brain health" presents and summarizes a good part of the literature discussing the impact of diet on the development of some age-related diseases. The topic is certainly of interest, and despite it is not novel and many other reviews are present on this research area, a nice update is always of interest, especially when it is well done as in this case.

Nevertheless, I feel that the authors completely did not present any information regarding the importance of hydration and proper fluid intake (they just mention a 2 L recommendation). I feel that the review would strongly benefit from a chapter related to the importance of healthy hydration for the elderly, as it should be adapted considering the physiological ageing process (Masot et al., 2020), and proper fluid intake might be associated with improved health and reduced risk of diseases (De Cavanagh et al., 2011; Pross N, 2017; Thornton S, 2011; Buoite Stella et al., 2021; Watso and Farquhar, 2019).

Author Response

Reviewer 2

The narrative review "Nutrition strategy for healthy aging: from prevention of geriatric diseases to promotion of cerebrovascular and brain health" presents and summarizes a good part of the literature discussing the impact of diet on the development of some age-related diseases. The topic is certainly of interest, and despite it is not novel and many other reviews are present on this research area, a nice update is always of interest, especially when it is well done as in this case.

Nevertheless, I feel that the authors completely did not present any information regarding the importance of hydration and proper fluid intake (they just mention a 2 L recommendation). I feel that the review would strongly benefit from a chapter related to the importance of healthy hydration for the elderly, as it should be adapted considering the physiological ageing process (Masot et al., 2020), and proper fluid intake might be associated with improved health and reduced risk of diseases (De Cavanagh et al., 2011; Pross N, 2017; Thornton S, 2011; Buoite Stella et al., 2021; Watso and Farquhar, 2019).

Reply: Thank you for the Reviewer's comment on the importance of adequate fluid intake for healthy ageing, and the manuscript has been amended. Thank you very much!

Thank You very much for the work and help of the Reviewers, we feel that improving the manuscript has greatly improved the scientific quality of the article. 

Sincerely:

Monika Fekete 

Round 2

Reviewer 1 Report

The authors addressed some of the suggestions. However, the manuscript still needs restructuring and improvements to be considered for publishing in Nutrients. Please see my suggestions:

- the title was changed but in my view the first option was better

- there are still too many in vivo studies cited for a study trying to address aging in human population

- all authors (not only one or two) should read the reviewers’ suggestions, and then respond and agree with the changes made

- there are over 300 citations, but several sentences need references

- Table 1 is not very clear - in the source you cite (Berendsen et al.), there are two separate tables (3 and 4) that make sense, but when they are combined the information is unclear. Anyway, these tables are built on rather old data, going back to 1991,1992, 1993, still recommending margarine... I suggest deleting this table; there are newer data in tables 4, 5, 7

- why make “Diets for healthy aging” (subsection 3.2.) a separate subchapter; I recommend including it in the beginning of chapter 3 (that was my suggestion in the first review), integrate it, eliminate the old info, and update the chapter

- Lines 309-313: in Ref. #193, is it 15 g/day linoleic acid? You say “There are no universal guidelines for the recommended ratio of omega-6 to omega-3 fatty acids for the prevention of cardiovascular diseases, but some reports suggest that a pro-portion of 1 to 4 is ideal (194).” That means omega-6 to omega-3 = 1 to 4!! It should be 4 to 1!

Lines 342-425: please concentrate these ideas; too much info

Section 7: there is only one mention about vitamin K, even though new evidence shows that vitamin K is a key factor in age-related disease prevention. Vitamin K has function in bone metabolism via the carboxylation of osteocalcin, a protein that transport and deposit calcium in bone (vitamin D, along with vitamin K and magnesium supplementations, can be a better strategy for reducing bone fractures in elderly); vitamin K activates matrix Gla protein and blocks the accumulation of calcium in the walls of blood vessels, thus making vitamin K safe approach for reducing cardiovascular disease morbidity and mortality.

Reference: the whole section should be rechecked - there are references with first names instead of last ones (see 144), incomplete data, some are cited twice